# Bridging Genotype to Phenotype in *KMT5B*-Related Syndrome: Evidence from RNA-Seq, ^18^FDG-PET, Clinical Deep Phenotyping in Two New Cases, and a Literature Review

**DOI:** 10.3390/genes16101174

**Published:** 2025-10-09

**Authors:** Davide Politano, Renato Borgatti, Giulia Borgonovi, Angelina Cistaro, Cesare Danesino, Piercarlo Fania, Gaia Garghetti, Andrea Guala, Isabella Orlando, Irene Giovanna Schiera, Claudia Scotti, Fabio Sirchia, Romina Romaniello, Gaia Visani, Denise Vurchio, Simona Mellone, Mara Giordano

**Affiliations:** 1Department of Brain and Behavioral Sciences, University of Pavia, 27100 Pavia, Italy; renato.borgatti@mondino.it (R.B.); isabella.orlando01@universitadipavia.it (I.O.); 2Department of Child Neurology and Psychiatry, IRCCS Mondino Foundation, Via Mondino 2, 27100 Pavia, Italy; gaia.garghetti@mondino.it (G.G.); rromaniello@dongnocchi.it (R.R.); 3Unit of Genetics, Clinical Biochemistry, University Hospital “Maggiore della Carità”, 28100 Novara, Italy; giulia.borgonovi@uniupo.it (G.B.); denise.vurchio@uniupo.it (D.V.); simona.mellone@uniupo.it (S.M.); mara.giordano@med.uniupo.it (M.G.); 4Department of Health Sciences, Università del Piemonte Orientale, 28100 Novara, Italy; 5AIMN Pediatric Study Group, 20146 Milan, Italy; angelinacistaro06@gmail.com; 6Nuclear Medicine Division, Salus Alliance Medical, 16151 Genoa, Italy; 7A.B.C. Associazione Bambini Cri du Chat Scientific Committee, 50126 Firenze, Italy; cesare.danesino@unipv.it (C.D.); gualaandrea0@gmail.com (A.G.); schierairene85@gmail.com (I.G.S.); 8Department of Molecular Medicine, University of Pavia, 27100 Pavia, Italy; fabio.sirchia@unipv.it (F.S.); gaia.visani01@universitadipavia.it (G.V.); 9Independent Researcher, 10100 Turin, Italy; pc.fania@gmail.com; 10SOC Pediatria, Ospedale Castelli, Verbania, 28922 Verbania, Italy; 11Independent Researcher, 90145 Palermo, Italy; 12Unit of Immunology and General Pathology, Department of Molecular Medicine, University of Pavia, 27100 Pavia, Italy; claudia.scotti@unipv.it; 13Medical Genetics Unit, IRCCS San Matteo Foundation, 27100 Pavia, Italy

**Keywords:** *KMT5B*, RNA-seq, neurodevelopmental disorder, autism spectrum disorder (ASD), Positron Emission Tomography (PET), neuropsychological assessment, genotype–phenotype correlation

## Abstract

**Background:** Autosomal dominant intellectual developmental disorder 51 (MIM #617788) is caused by pathogenic variants in KMT5B, a histone methyltransferase essential for transcriptional repression and central nervous system development. The disorder manifests as a complex neurodevelopmental syndrome with variable neurological and systemic features. **Methods:** Two adolescents with nonsense KMT5B variants underwent detailed clinical, neuropsychological, and neuroimaging evaluations, including MRI and ^18^FDG PET/CT, analyzed with Statistical Parametric Mapping against matched controls. RNA sequencing was performed, and the literature was reviewed to assess genotype–phenotype correlations. **Results:** Both patients showed global developmental delay, progressing to autism spectrum disorder (ASD) and developmental coordination disorder (DCD), without intellectual disability (ID). The MRI was normal, but neuropsychological testing revealed executive function impairment, expressive language deficits, and behavioral disturbances. PET/CT consistently demonstrated cerebellar and temporal lobe hypometabolism, correlating with symptom severity. RNA sequencing identified shared dysregulated pathways, notably DDIT4 upregulation, linked to synaptic dysfunction and neuronal atrophy in animal models. **Conclusions:** The findings highlight cerebellar involvement in DCD and ASD, medial temporal lobe contribution to ASD and executive dysfunction, and DDIT4 as a possible molecular signature of KMT5B loss-of-function. An integrative multimodal approach refined genotype–phenotype correlations and revealed novel brain regions and pathways implicated in KMT5B-related disorders.

## 1. Introduction

Autosomal dominant intellectual developmental disorder 51 (#MIM 617788) is a complex neurodevelopmental disorder (NDD) caused by pathogenic variants in the lysine methyltransferase 5B gene (*KMT5B*). The first association between *KMT5B* pathogenic variants and disease was reported in 2017 by Stessman and colleagues. They described seven patients with NDD, mainly global developmental delay (GDD) and autism spectrum disorder (ASD), who carried both missense and likely gene disruptive (LGD) variants in *KMT5B,* mostly de novo in origin. This correlation was further supported by experiments in Drosophila, where knockdown of *Hmt4-20*, the fly ortholog of human *KMT5B*, resulted in habituation learning deficits [1].

After the first description, many patients have been reported, expanding the phenotypic spectrum to include non-specific facial and body dysmorphisms, ligamentous laxity, other NDDs such as isolated speech delay/disorder and Tourette syndrome, behavioral/psychiatric disorders, and congenital malformations, mostly confined to the cardiovascular and genitourinary systems, with other sporadic findings. Brain MRI findings were diverse and nonspecific [2,3,4,5,6,7,8,9].

In humans, *KMT5B* (*lysine methyltransferase 5B*) is a gene located on chromosome 11q13.2, encoding a histone methyltransferase that methylates lysine 20 of histone H4 (H4K20), producing H4K20me2 and H4K20me3 [10]. These modifications regulate transcription and genome integrity [10,11]. H4K20me3 is a marker for transcriptional repression and, along with H4K20me2, promotes TP53BP1 foci formation in DNA damage response and non-homologous end-joining (NHEJ) repair in nonreplicating neurons [11]. *KMT5B* deficiency leads to increased p53 expression, also observed in brains of autistic subjects [12,13], and loss of H4K20me3, potentially causing gene upregulation. RNA sequencing using prefrontal cortex samples of mice undergoing knockdown of *kmt5b* by Wang et al. [13] revealed that differentially expressed genes—linked to protein folding, translation, chromatin modification, and ubiquitin-mediated degradation—are upregulated, likely contributing to ASD-related synaptic and behavioral abnormalities.

In particular, a significant upregulation of p53 downstream target genes was observed in mice, notably including *DDIT4* (also known as *REDD1*). This gene is critically involved in mediating synaptic dysfunction and loss following chronic stress. Previous studies [14] have shown that DDIT4 is strongly induced in response to prolonged stress exposure and contributes to impaired synaptic plasticity, likely through suppression of the mTOR (mechanistic Target Of Rapamycin) signaling pathway.

The neuropsychological and clinical description of individuals with *KMT5B* pathogenic variants has primarily focused on understanding the developmental challenges associated with this genetic condition. However, no detailed neuropsychological assessments targeting specific brain functions have been conducted in this syndrome. Cognitive, behavioral, and emotional aspects can be assessed only through neuropsychological evaluation. Beyond global cognitive functioning, specific cognitive domains can be thoroughly examined, including attention and executive functions—conceptualized as a set of top-down cognitive processes (inhibition, working memory, and shifting) that form the foundation for higher-order functions such as reasoning, problem-solving, and planning [15]—as well as memory, social perception, sensorimotor functions, and language. Additionally, questionnaires facilitate the evaluation of social, emotional, and behavioral functioning. The integration of results from these assessments provides a comprehensive neuropsychological evaluation, essential for a detailed characterization of functional impairments, useful for monitoring developmental progress, and tailoring individualized interventions.

It is well established that ^18^F-FDG PET/CT is an important tool for the assessment of cerebral metabolism in several neurologic diseases [16,17,18]. However, most of the studies that have been carried out using brain PET/CT in rare genetic neuropediatric conditions have been targeted to better understand etiology of movement and progressive disorders [19,20,21], with only few articles describing ^18^F-FDG PET/CT brain findings in patients with a genetically confirmed neurodevelopmental disorder [22,23,24,25]. Nonetheless, abnormalities in glucose brain metabolism detected by PET have been extensively studied in ASD: hypometabolic areas are localized in the temporal lobe, prevalently in its medial portion (i.e., amygdala, hippocampus, and parahippocampal gyrus involvement), and in other temporal regions, and hypermetabolic areas are localized prevalently in the frontal lobes, although for both alterations many areas seem to be altered, resulting in poor specificity [26,27].

In the present study, our aim was to make a critical review of the literature regarding clinical and radiological presentation of the disease, and to integrate these data with ^18^F-FDG PET/CT scan results and deep neuropsychological phenotyping of two adolescent female patients both showing GDD in the first years of life and later diagnosed with ASD and DCD. RNA sequencing (RNA-seq) was also performed on peripheral blood samples from the two patients to investigate potential differences in gene expression and dysregulated pathways resulting from loss-of-function mutations in *KMT5B.* Both patients carry two novel de novo pathogenic variants in *KMT5B*, thus allowing us to describe similarities and differences in functioning both at the clinical and neurometabolic level.

## 2. Materials and Methods

### 2.1. Clinical Data

Clinical data were obtained from medical records and by standardized clinical testing. A comprehensive clinical assessment was conducted, including general and neurological examinations, targeted dysmorphological evaluation, and neuropsychological testing. Age-specific neuropsychological assessment batteries were administered to evaluate various cognitive functions, as well as emotional and behavioral aspects, to establish the participants’ neuropsychological profiles. The Wechsler Intelligence Scale for Children—Fourth Edition [28] was employed to assess global cognitive functioning, derive Full Scale Intelligence Quotient (FSIQ), and measure distinct aspects of intelligence. Specific cognitive domains were thoroughly examined. Attention and executive functions [15] were assessed using TCM (Test delle Campanelle Modificato) [29,30], NEPSY-II Battery (Auditory and Visual Attention and Inhibition) [31], BVN 12–18 Battery (Forward Verbal Digit Span, Backward Verbal Digit Span, Corsi Block-Tapping Test) [32], MEA Battery (Alpha Span) [33], and TOL Tower of London [34]. Additionally, memory abilities were assessed using List Memory and Memory for Design from the NEPSY-II Battery. Social perception was evaluated through Affect Recognition and Theory of Mind from the NEPSY-II Battery. Sensorimotor functions were examined using Imitating Hand Positions and Manual Motor Sequences from the NEPSY-II Battery, along with the VMI Visual-Motor Integration test [35]. Language abilities were assessed through Speech Discrimination, Lexical Denomination, and Sentence Generation from the BVN 12–18 Battery. Finally, emotional and behavioral difficulties were explored using Child Behavior Checklist for ages 6–18 (CBCL 6–18) [36], Social Communication Questionnaire—Lifetime Version (SCQ-V) [37], and Conners 3 [38]. The integration of the results from these assessments provides a comprehensive neuropsychological evaluation, essential for a detailed characterization of functional impairments.

### 2.2. Neuroradiological Data

MRI scans were performed on a 1.5 T GE HDxt scanner (General Electric Healthcare, Waukesha, WI, USA) in both patients.

Brain PET/CT was performed after both patients fasted for at least 6 h, and glycemia was measured. Patients were placed in a quiet, darkened room at least twenty minutes before the administration of ^18^F-FDG and remained there throughout the uptake phase. They received 185 MBq of ^18^F-FDG through an intravenous cannula according to the European Association of Nuclear Medicine procedural guidelines [39]. PET/CT acquisition commenced 60 min after tracer injection and lasted about 15 min. Head motion was minimized using a polycarbonate head holder. Scans were performed on an Ingenuity TF PET/CT system (Philips Healthcare, Cleveland, OH, USA) in three-dimensional mode, with a 30 cm axial field of view and no interplane gap. ^18^F-FDG PET/CT brain data were collected through sequential scans: CT data were used for attenuation correction of the PET data, with a thickness of 3.75 mm, 140 kV, and 60–80 mA/s. PET data were reconstructed using three-dimensional iterative ordered-subsets expectation maximization (OSEM) with corrections for randoms, scatter, and attenuation (VUE-point; 28 subsets, 4 iterations). Images were generated in 256 × 256 matrices with a reconstructed voxel size of 1.33 × 1.33 × 2.00 mm.

### 2.3. Statistical Analysis

The PET/CT findings were compared with an age-matched control group drawn from a cohort of 40 individuals previously included in other research studies [24,25,40], using statistical parametric mapping (SPM, version 12; Wellcome Department of Cognitive Neurology) implemented in MATLAB (version 9.4; MathWorks), as described in previous studies [24,25]. The SPM coordinates were adjusted to match Talairach coordinates using a subroutine implemented by Matthew Brett (http://brainmap.org/index.html accessed on the 25 February 2025). Brodmann areas (BAs) were then identified within 0–3 mm of the corrected Talairach coordinates of the SPM output isocenters, after importing the corrected coordinates via the Talairach client (http://www.talairach.org/index.html accessed on the 25 February 2025) in SPM. One-way ANOVA was conducted using the SPM f-contrast routine to compare the patients and control groups. A two-sample unpaired *t*-test was applied to identify regions of relative hypo- and hypermetabolism in individual patients. Age and sex were included as nuisance variables. A *p* value threshold of 0.05, corrected for multiple comparisons using the false-discovery-rate option, was applied to explore SPM t-maps at both cluster and voxel levels, considering clusters larger than 100 voxels as significant. If statistical significance was not achieved, voxel-level thresholds were further explored at *p* < 0.001, uncorrected for multiple comparisons.

### 2.4. NGS Targeted Sequencing and Sanger Validation

A targeted gene panel, including genes correlated to NDDs, was used to sequence the DNA of patient 1. DNA libraries were prepared using SureSelect Design ID: S3442724 in combination with the Agilent Magnis NGS Prep system (Agilent Technologies, Santa Clara, CA, USA), following the protocol for Illumina Multiplex Sequencing as previously reported [41]. Pooled samples were analyzed simultaneously on the Nextseq 1000 platform (Illumina, Inc., San Diego, CA, USA) using a NextSeq™ 1000/2000 P2 Reagent Cartridge (2 × 150 bp cycles, Illumina, San Diego, CA, USA), achieving an estimated coverage of 200X. For each sample, more than 98% of the target gene sequence was covered by at least 50 reads. Variant calling was performed using the enGenome eVai software (evai.engenome.com accessed on the 17 Novemeber 2024) [42].

Whole exome sequencing (WES) was performed on patient 2 using genomic DNA with the Twist Human Core Kit (Twist Bioscience) on a NovaSeq6000 sequencer (Illumina). Reads were aligned to the human reference genome (GRCh37) using BWA v0.7.5, and variants were called with GATK Unified Genotyper.

For both patients, variant validation was performed using Sanger sequencing. Primers were designed with Primer3 v.0.4.0 (https://primer3.ut.ee/ accessed on the 15 December 2024), and the resulting amplicons were purified using ExoSAP-IT™ Express PCR Product Cleanup Reagent (Applied Biosystems, Foster City, CA, USA). Sequencing was carried out in both directions with the BigDye Terminator v1.1 Cycle Sequencing Kit (Applied Biosystems) and analyzed on a SeqStudio Genetic Analyzer (Applied Biosystems).

Variants were classified using the American College of Human Society (ACMG) guidelines [43].

### 2.5. RNA Sequencing

Total RNA was extracted from peripheral blood mononuclear cells (PBMCs), isolated from whole blood using Lympholyte^®^ (Cedarlane, Burlington, NC, USA), with the miRNeasy Tissue/Cells Advanced Mini Kit (QIAGEN, Hilden, Germany). Samples were obtained from two patients and two age- and sex-matched controls, and each was processed in duplicate. RNA concentration and purity were measured using NanoDrop One (Thermo Fisher Scientific, Waltham, MA, USA), while RNA integrity was assessed with the RNA ScreenTape Assay (Agilent Technologies) on the TapeStation 4150, yielding RNA integrity numbers. For both patients, RNA sequencing was carried out using the SureSelect XT HS2 mRNA Library Preparation System (Agilent Technologies) in combination with the Agilent Magnis NGS Prep system, following the manufacturer’s protocol with 500 ng of RNA per sample. Library quality was evaluated using the D1000 ScreenTape Assay on the TapeStation 4150 (Agilent Technologies). Sequencing was performed on a NextSeq 1000 platform (Illumina), analyzing four samples in parallel (two patients and two controls) with a NextSeq™ 1000/2000 P2 Reagent Cartridge (300 Cycles, Illumina), yielding more than 40 million unique reads per sample. Data were analyzed with the Dynamic Read Analysis for GENomics (DRAGEN) RNA v.3.8.4 pipeline (Illumina).

### 2.6. Literature Review

An updated literature review as of 9 March 2025, was conducted and reported in Table 1 and Appendix A, including all articles that provided a detailed characterization of patients, using the following search string: “KMT5B” AND “syndrome” AND “neurodevelopment”. The article by Shiota and colleagues [44] was not included because it did not include extensive clinical data about patients.

## 3. Results

### 3.1. Patients’ Clinical-Radiological Findings

Patient 1 was a 13-year-old girl, firstborn of non-consanguineous, healthy parents. There was no family history of neurological, psychiatric, or other potentially genetic conditions. Pregnancy was complicated by intrauterine growth restriction (IUGR) from the 28th week of gestation. Delivery was by urgent cesarean section due to fetal distress at 34 weeks and 6 days of gestational age. At birth, her growth parameters were within normal limits (weight 1825 g, length 46 cm, and cranial circumference 30 cm), and Apgar scores were 9 and 10 at 1 and 5 min, respectively.

She exhibited global developmental delay, including mild motor delay (sitting at 9 months), fine and gross motor incoordination, and delays in social and communicative skills. Her first words emerged around 3 years, and she began forming phrases after age 5. From three years of age, the patient experienced sleep disturbances, including sleep-onset insomnia, episodes of night terrors, and sleepwalking. She received a diagnosis of ASD at age 11. An NGS panel for NDDs detected a de novo heterozygous likely pathogenic variant: (PVS1, PM2) *KMT5B* (NM_017635.5): c.2495_2497del (p.Ser832Ter).

On the most recent examination, at 13 years of age, notable findings included macrocephaly, hypotonia, gross and fine-motor impairment leading to the diagnosis of DCD, reduced expressive vocabulary, and a pragmatic language disorder. She also displayed restricted, repetitive behaviors, interests, and activities that, combined with language and socialization difficulties, led to ASD diagnosis. Cognitive evaluation was within normal limits (IQ 104, VCI 108, PRI 102, WMI 100, PSI 100). Mild executive function impairments were associated with learning difficulties. ADHD traits were also observed. Behavioral and emotional difficulties emerged from standardized questionnaires although not clinically relevant (see Table 2). Brain MRI was unremarkable.

SPM analysis on ^18^F-FDG distribution revealed a statistically significant hypometabolism in the left fusiform gyrus and uncus (BAs 37, 28), in the bilateral anterior (culmen), and the posterior cerebellar lobes (bilateral declive; pyramis and inferior semi-lunar lobule on the left; tonsil on the right) compared to the control cohort (*p* < 0.001). Furthermore, a trend of hypometabolism was also detected in the bilateral frontal paracentral lobules (BAs 6), the right inferior temporal and fusiform gyri (BA 20 and hippocampus), the left superior and inferior temporal gyri (BAs 38,20) and uncus, and the bilateral precuneus and cuneus (BAs 7; right Bas 17,18,19,23; and left BA 18).

No statistically significant areas of relative hypermetabolism were found at *p* < 0.001. A trend was found in the left frontal superior, middle, and medial frontal gyri (BAs 9, 10) (*p* = 0.01).

Patient 2 was a 13-year-old girl, the third born to non-consanguineous, healthy parents. Family history was negative for neurological, psychiatric, or other plausibly genetic conditions. The pregnancy was uneventful, although the mother reported using estrogen pills during the first trimester. Delivery was via cesarean section due to a previous cesarean section at 39 weeks of gestational age. Auxological parameters were within normal range. She later exhibited global developmental delay, achieving head control at five months, independent sitting at eight months, and autonomous walking at two years. Language development was delayed, with vocalizations and babbling persisting until four years of age, followed by the later acquisition of more functional language. She also demonstrated difficulties in both gross and fine motor coordination and exhibited toe-walking until the age of three. The patient experienced sleep disturbances, including sleep-onset and maintenance insomnia, sleepwalking, and episodes of night terrors, which have recently decreased in frequency. The patient underwent a WES analysis for GDD and was found to harbor the novel heterozygous de novo pathogenic (PVS1, PM2, PP5) *KMT5B*(NM_017635.5): c.559C > T; (p.Arg187Ter) variant.

On the most recent examination, at 13 years of age, notable findings included macrocephaly, hypertelorism, downslanting palpebral fissures, horizontal eyebrows, a bulbous nasal tip, a wide nasal bridge, diastema of the upper central incisors, gingival hypoplasia, long and slender fingers, distal ligamentous laxity, scoliosis, bilateral pes planus, and obesity with signs of metabolic syndrome (including insulin resistance and reduced HDL cholesterol) (see Figure 1).

The patient presented with a language disorder characterized by speech dyspraxia, reduced receptive and expressive vocabulary, and a pragmatic language disorder. She also exhibited restricted, repetitive patterns of behavior, interests, and activities, which, in conjunction with her language and socialization difficulties, led to a diagnosis of ASD. A distinctive feature of her condition is unusual social behavior, notably an absence of stranger danger. She also exhibited gross and fine-motor impairment leading to the diagnosis of DCD. The patient also showed attention deficit hyperactivity disorder (ADHD) traits.

Cognitive evaluation was within normal limits, with a total IQ quotient of 96 (Verbal Comprehension Index (VCI) 100; Perceptual reasoning Index (PRI) 119; Working Memory Index (WMI) 76; Processing Speed Index (PSI) 82). Executive functions were impaired thus leading to some learning difficulties. Behavioral and emotional difficulties emerged from standardized questionnaires (see Table 2). Brain MRI was within normal limits. SPM analysis of metabolic brain study revealed a statistically significant relative hypometabolism in the left uncus (BA 20) and left parahippocampal gyrus (or hippocampal gyrus), and in the bilateral posterior cerebellar lobes (pyramis, semilunar lobule, and tonsil) compared to the control cohort (*p* < 0.001). Furthermore, a trend of hypometabolism was also detected in the bilateral parahippocampal gyri (BAs 28, 36), the left medial and inferior frontal gyri (BAs 25, 47), the superior, middle, inferior, and fusiform temporal gyri and uncus (BAs 38, 20), the anterior cingulate gyrus (BA 32), the right hippocampus, the bilateral posterior cerebellar lobes (declive), and the anterior cerebellar lobes (bilateral culmen and right lingual lobule, *p* < 0.01).

Statistically significant relative hypermetabolism (*p* < 0.001) was found in the right inferior parietal gyrus (BA 40) and postcentral gyrus (BAs 2, 3, Figure 2).

### 3.2. Differential Gene Expression

RNA sequencing was conducted on peripheral blood samples from the two patients carrying *KMT5B* pathogenic variants and two sex- and age-matched healthy controls to investigate transcriptomic alterations associated with *KMT5B* deficiency. As a first step, we assessed *KMT5B* expression levels. Both patients exhibited significantly reduced *KMT5B* expression compared to controls: patient #1 showed a log_2_ fold change in –0.50 (adjusted *p* = 0.00387), and patient #2 a log_2_ fold change in –0.60 (adjusted *p* = 0.00030). This downregulation might be consistent with nonsense-mediated mRNA decay (NMD) [5,14,45,46]. However, a closer inspection of the RNA-seq alignment data revealed that transcripts from the mutant allele were still detectable in both patients, accounting for 37% of total *KMT5B* reads in patient #1 and 36% in patient #2. This is expected for the *KMT5B: c.2495_2497del* variant, as the deletion occurs in the last exon of the gene, a region typically resistant to NMD. Therefore, the detectable levels of the mutant transcript likely indicate incomplete degradation or escape from NMD, potentially allowing for the synthesis of truncated protein products.

We next investigated whether genes previously reported to be dysregulated in the prefrontal cortex of *KMT5B* knockdown C57BL/6J mouse models [13] exhibited similar expression patterns in our patients, thereby reinforcing the relevance of these pathways in the disease mechanism. Specifically, *KMT5B* deficiency led to a significant upregulation of Ddit4 expression at both the mRNA and protein levels in the murine model, attributable to activation of the p53-mediated DNA damage response pathway, similarly to what observed in postmortem PFC tissue from individuals with ASD [13]. Notably, *DDIT4* (also known as *REDD1*) was significantly upregulated in both patients: in patient#1, *DDIT4* showed a striking log_2_ fold change of 180.89 (adjusted *p* = 2.39 × 10^−10^), whereas in patient #2, the log_2_ fold change was 0.69 (adjusted *p* = 0.00038). This finding is particularly relevant, as DDIT4 upregulation has been implicated in synaptic dysfunction and neuronal atrophy associated with chronic stress exposure [14].

To further explore the molecular consequences of *KMT5B* haploinsufficiency, we performed a pathway enrichment analysis through Metascape (https://metascape.org/gp/#/main/step1 on the 30 November 2024) considering the differentially expressed genes (DEGs). DEGs were defined using the following criteria: padj < 0.05 and log2FoldChange > 1 for upregulated genes, and log2FoldChange < −1 for downregulated genes. Applying these criteria, we identified 384 overexpressed genes in patient #1 and 629 in patient #2. Notably, 157 genes were significantly upregulated in both patients and were associated with enriched biological pathways (Appendix A). Several pathways were also found to be enriched in patients in a previous study [5]. Among these, the kinase and transcription factor activation pathways showed the most elevated signal (*p* = 2.24 × 10^−7^), including the TGF-β signaling pathway (*p* = 5.75 × 10^−4^). A significant enrichment of pathways involved in the positive regulation of nervous system development was also observed (GO:0051962, *p* = 6.31 × 10^−4^), driven by genes such as *CRABP2, EPHB2, VEGFA, NUMBL, CLCF1,* and *PARP6* (Appendix A). Furthermore, the pathway associated with striated muscle cell development (*p* = 8.51 × 10^−4^) was also significantly enriched, with the contributions of *CAPN3, CFL2*, *MYBPC3*, *VEGFA*, *WNT10B*, *LIMCH1*, *PMP22*, *FBLN2*, and *SPTBN5* (Appendix A). Interestingly, we also identified enrichment in pathways that were not previously reported in humans, as the regulation of ubiquitin-dependent protein catabolic process (GO:0032434, *p*-value = 2.69 × 10^−2^), which was previously reported only in mice studies. Among the genes with decreased expression, 528 were identified in patient #1 and 494 in patient #2. Of these, 230 genes were shared between both patients and were significantly enriched in pathways related to the defense response to viruses (*p* = 5.89 × 10^−15^) and regulation of nuclear division (*p* = 7.08 × 10^−12^) (Appendix A).

### 3.3. Structural Effects of Mutations

The *KMT5B* gene encodes an 865 amino acid protein and includes a SET domain (residues 193-308, Prosite) and five intrinsically disordered regions (residues 1-62, 363-444, 590-655, 716-740 and 816-850, MobiDB) mainly located in the C terminal domain. Understandably, the KMT5B protein structure predicted with AlphaFold3 (https://alphafold.ebi.ac.uk/entry/Q4FZB7 accessed on 30 November 2024) shows these regions with pLDDT < 50, indicating low reliability of the model and making a detailed analysis difficult. In contrast, a detailed description of a cryoEM structure of residues 1-390 (with a K20M mutation introduced to stabilize the complex), comprising the SET domain, is available (PDB ID: 8JHG, Figure 3) [47].

From this structure, it can be noticed that the N-terminal region of KMT5b makes extensive contacts with three elements of the nucleosome: the DNA, the H4 tail, and the H2A-H2B acidic patch, a region which normally interacts with the N-terminal domain of H4 to create supercoiled chromatin. Particularly, the H4 N-terminal tail, which contains the target K20 residue, is positioned within the hydrophobic pocket of KMT5b, ready for its methylation. There are many critical residues located in this domain: R220, R286, R257, G285, K258, S283, S255, Y349, R352, and R357, each involved in either binding or catalytic activity [47].

According to NGS data, variant c.2495_2497del; (p.Ser832Ter) in patient 1 might allow production of an almost complete protein (832 residues out of 885). Particularly, the resulting peptide, if functionally folded, would contain all the elements necessary for binding and for catalysis described above, but it would lack, for example, residue S832, a potential phosphorylation site (as predicted by NetPhos—3.1). Moreover, the last 91 residues of KMT5b have been shown to be essential for interaction with Glutamate receptor-interacting protein 1 (GRIP1) [48], indicating another example of potential functional dysregulation consequent to the mutational event.

Variant c.559C > T; (p.Arg187Ter) in patient 2, a stop gained variant with a CADD score of 33, leads to a much earlier premature truncation than the one in patient 1, leading to the loss of the SET domain residues of the binding pocket hosting the H4 N-terminal tail and those needed to bind the acidic patch of the nucleosome. In this case, interaction with the nucleosome might be maintained, but the catalytic activity would be totally lost.

## 4. Discussion

The *KMT5B* gene encodes an 865 amino acid-protein relevant in regulating transcription of different genes important for neural development [5]. It contains the SET domain (aa193-aa308), through which it catalyzes histone methylation. However, other regions of the protein, even if not molecularly characterized due to their intrinsic characteristics [5], are targeted by missense variants and show important functions such as protein–protein interactions with other transcriptional regulators essential for complete function of the protein [49]. In both our patients, the identified variants introduce a premature termination codon, although patient 2 variant is far more proximal to the N-terminal than the variant displayed by patient 1.

All variants described so far likely result in haploinsufficiency, as suggested by the high number of patients with protein-truncating variants (PTVs), the lack of specific genotype–phenotype correlations among the different types of described variants, the effect of knockout of *kmt5b* in mice recapitulating human disease, and the presence of chromosomal deletions encompassing KMT5B—without other relevant disease-associated genes—exhibiting a phenotype consistent with KMT5B-related neurodevelopmental syndrome, as described by Faundes and colleagues in 2017 [3].

To better understand the molecular mechanisms underlying the clinical phenotype and to assess the functional consequences of KMT5B haploinsufficiency, we performed transcriptome analysis through RNA-seq on peripheral blood samples. This approach allowed the identification, alongside the wild-type form, of the transcript variants present in significant amounts. It is naturally tempting to associate the molecular effect of the longer transcript of patient 1 with the less severe phenotype: from the structural point of view, in fact, the protein might be almost fully functional, except for the missing regulatory sites. In contrast, the shorter transcript of patient 2 could lead not only to haploinsufficiency but also to the generation of a natural and potent antagonist of the wild-type counterpart able to compete with its binding site and preventing its function. In this respect, transcriptome analysis also aimed to identify differentially expressed genes and dysregulated biological pathways that may contribute to the pathogenesis observed in patients carrying loss-of-function variants in *KMT5B.* While peripheral blood RNA-sequencing provides a minimally invasive approach to explore gene expression alterations, it is important to acknowledge its limitations in the context of neurodevelopmental disorders. The transcriptomic landscape of blood may not fully reflect the molecular changes occurring in the brain, particularly given the tissue-specific nature of gene regulation. Nonetheless, it remains a practical and informative approach for identifying systemic molecular signatures relevant to neurodevelopmental disorders, especially when integrated with complementary data on animal models and data on other patients. KMT5B is a histone methyltransferase that catalyzes the demethylation of histone H4 at lysine 20 (H4K20me2). This chromatin change relaxes normally compacted regions, alters gene expression, and impairs DNA repair processes. The most relevant dysregulated pathway in the here described patients involved kinases and transcription factor activation. This is particularly relevant, as kinase signaling cascades and transcription factor networks are fundamental regulators of neurodevelopmental processes, including neuronal proliferation, migration, differentiation, and synaptogenesis [7], and their dysregulation may play a pivotal role in the pathophysiology associated with KMT5B haploinsufficiency.

Loss of KMT5B function impairs DNA repair by reducing histone H4K20 di- and trimethylation (H4K20me2/3), which are essential for recruiting the DNA repair protein 53BP1 (p53 binding protein 1) to sites of double-strand breaks. Without proper 53BP1 recruitment, DNA repair, particularly via non-homologous end joining, is compromised, leading to genomic instability, and activated p53 regulates genes involved in stress responses, including its downstream target *DDIT4/REDD1* whose upregulation is associated with synaptic dysfunction and loss under chronic stress [4]. While this can be adaptive in eliminating damaged cells, excessive or prolonged DDIT4 activation contributes directly to neuronal loss and neurodegeneration. Notably, the here described patients showed in their blood RNA a high level of DDIT4 transcript in line with what was observed in mice [13]. Moreover, excessive DDIT4 expression has been shown to be detrimental to neurodevelopment, and its expression has been correlated to pathophysiology of autism spectrum disorder (ASD) [50]. In this context, sustained DDIT4 activation may disrupt key signaling pathways, including mTOR, thereby affecting neuronal survival, synaptic development, and overall brain maturation. It has been recently observed that DDIT4 upregulation contributes to autism-like behaviors via a novel ferroptosis mechanism [51]. The convergence of findings from animal models to patient-derived data, including ours, suggests that increased DDIT4 might be a recurring feature in at least a subset of cases with NDDs and, in particular, those carrying loss of function mutations in KMT5B, although evidence from more patients is necessary.

From a clinical standpoint, our most recent comprehensive review of the literature portrayed in Table 1 and Appendix A provides a detailed overview of *KMT5B*-associated disorder, reporting patients, mostly males, from 5 weeks to 50 years of age, with a median age of approximately 11 years. With regard to the male-to-female ratio, the excess of males is not statistically significant, but it may suggest a higher penetrance in males, which should be confirmed with a larger number of patients. Among the patients reported in the literature, the predominant neurodevelopmental trajectory involves GDD progressing to intellectual disability/learning difficulties, often in association with ASD. Macrocephaly is a recurrent feature, observed in the majority of cases, while other associated craniofacial and body dysmorphisms are nonspecific. Throughout the lifespan, approximately one-quarter of affected individuals experience behavioral and psychiatric disorders, ranging from self-injury and self-mutilation to psychosis. Some cases also present with a history of developmental regression or other sporadic neuropsychiatric manifestations. Epilepsy has been reported in twelve out of fifty-two patients (23%) for whom clinical data were available; however, due to the lack of detailed information, no specific electroencephalographic (EEG) patterns or seizure types could be inferred. Sleep disturbances are reported in 12% of patients (seven patients), with a prevalence of sleepwalking (three out of the six patients for whom sleeping disorder was described thoroughly), although these disturbances are commonly underdiagnosed in complex genetic conditions [52]. Hypotonia is the primary finding during neurological examination, followed by deficits in motor coordination or DCD, another under-recognized disorder [53]. Brain MRI findings are nonspecific and sporadic; each observed in one or two patients. These include dilated ventricles, white matter non-specific abnormalities, not otherwise specified leukodystrophy, Chiari malformation, periventricular nodular heterotopia (PVNH), perisylvian polymicrogyria, cerebellar subdural hygroma, and corpus callosum dysgenesis. Additionally, congenital malformations are observed in approximately 17% of cases, predominantly affecting the cardiovascular and genitourinary systems; three patients showed metopic craniosynostosis [9]. Other recurrent but less specific clinical findings include ligamentous laxity, gastrointestinal symptoms such as constipation and gastroesophageal reflux disease, as well as other more sporadic manifestations [1,2,3,4,5,6,7,8,9].

Interestingly, our patients exhibit remarkably similar neuropsychological findings. Both were diagnosed with ASD and DCD while maintaining overall cognitive abilities within the normal range but showing some degree of learning difficulties. In the literature, only a minority of patients have been reported to exhibit normal intellectual functioning or isolated learning difficulties without intellectual disability, as observed in our patients. They indeed displayed abnormalities in executive functions, mild expressive language impairment, and internalizing emotional and behavioral difficulties, including withdrawal, fearfulness, anxiety, inhibition, and somatization. However, the second patient presented with a more severe phenotype, characterized by a significant reduction in sustained attention and greater impairment in memory compared to the first patient, who exhibited only a deficit in short-term visuospatial memory. Additionally, social perception and sensory-motor functions were more impaired in the second patient.

PET abnormalities were also strikingly similar between the two patients, with a prevalent hypometabolism in the temporal (mainly medial temporal) lobe and cerebellum in both patients. Hypometabolism in the medial temporal regions is a recurrent feature in autism as previously described (“Introduction” section), but these areas are also associated with executive functions and memory [54]. Hypometabolic changes in the cerebellum, the main organ for the coordination of body parts in movement [55], may explain the motor incoordination described in our patients and recurrent in KMT5B-related syndrome. However, hypometabolic changes have also been described in cerebella from autistic patients in two different studies [56,57]; indeed, the cerebellum has an important role in social cognition [58], and its alterations have been studied as a cause of autism for years [59]. Zebrafish and mouse studies [5] found out that KMT5B is highly expressed during central nervous system (CNS) embryonic development and later in post-natal life, particularly in certain areas among which hippocampus and cerebellum, areas where there is active neuronal proliferation also after birth and that are hypometabolic in our patients.

Hypermetabolic areas, significantly more relevant in patient 2, differ in the two patients. The significance of increased metabolic activity in certain brain regions remains uncertain. For example, in conditions like motor neuron disorders, such metabolic alterations have been associated with heightened astroglial responses, a key feature of immune system involvement in the central nervous system [24,60]. However, a prior investigation involving individuals with Cri-du-chat syndrome [24], another genetic NDD, identified hypermetabolic findings primarily linked to more pronounced clinical manifestations. This suggests that such alterations could contribute to phenotype determination in our two cases, given that the relevance of these metabolic changes appears greater in patient 2 whose clinical picture is more severe.

## 5. Conclusions

Future studies should aim to recruit a larger cohort of patients to strengthen our PET and RNA-seq findings and their correlation to neuropsychiatric symptoms and signs not only in KMT5B-related syndrome but also in other genetically confirmed complex NDD. Indeed, the main limitations of our study include the small sample size, due to the rarity of the disease, and the genetic heterogeneity between the two patients, despite both mutations likely resulting in a loss-of-function (LoF) effect. Nonetheless, the main concepts outlined in our study are:-The importance of comprehensive neuropsychological phenotyping to enhance correlations between phenotypic data, advanced imaging techniques, and genetic findings and to better guide families during re-habilitation.-The necessity of studying neurodevelopmental disorders (NDDs) using functional and metabolic approaches in order to deeply characterize patients.-Deep phenotyping of patients clustered by precise genetic rather than functional diagnosis (i.e., KMT5B-related disorder as opposed to ASD) leads to more reliable and trustable data.-In the absence of clear macroscopic brain alterations—primarily, but not exclusively—it is essential to assess metabolic brain changes to identify the possible primary areas affected by the disease.

## Figures and Tables

**Figure 1 genes-16-01174-f001:**
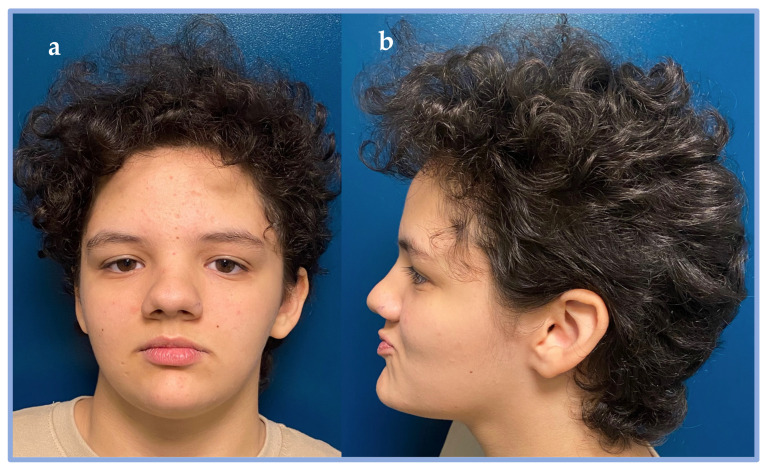
Dysmorphisms of patient 2. (**a**) Front view. (**b**) Side view. In the picture, the following dysmorphisms can be appreciated: macrocephaly, hypertelorism, downslanting palpebral fissures, horizontal eyebrows, a bulbous nasal tip, a wide nasal bridge.

**Figure 2 genes-16-01174-f002:**
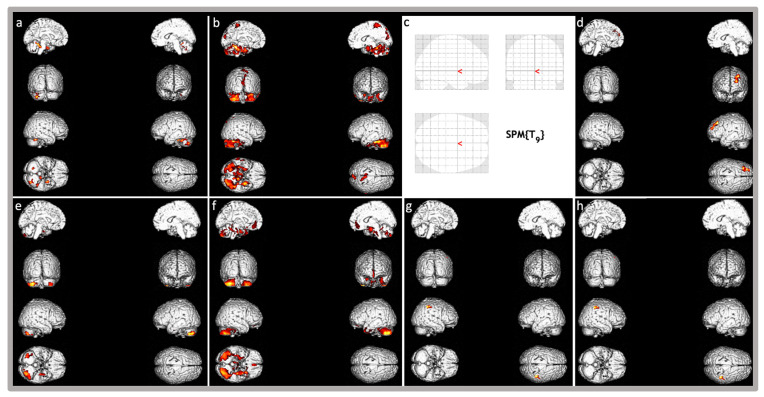
Brain three-dimensional rendering showing those regions in which ^18^F-FDG uptake was significantly different in the single patient with *KMT5B*-related disorder compared to the normal control group database. Images are color-graded from yellow to red, with red indicating stronger differences (higher z-values). Top images: (**a**,**b**) show relative hypometabolism in patient 1 (respectively, *p* < 0.001 and *p* < 0.01), (**c**,**d**) show relative hypermetabolism (*p* < 0.001 and *p* < 0.01, respectively). Of note, patient 1 does not have any relative hypermetabolic area at *p* < 0.001 as in Figure (**c**). Bottom images (Patient 2): (**e**,**f**) show relative hypometabolism (respectively, *p* < 0.001 and *p* < 0.01), (**g**,**h**) show relative hypermetabolism (respectively, *p* < 0.001 and *p* < 0.01).

**Figure 3 genes-16-01174-f003:**
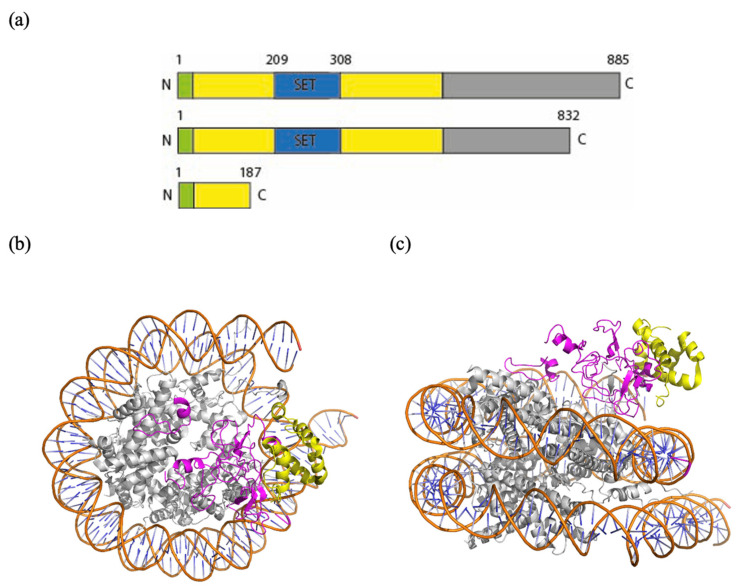
(**a**) Schematic view of the domain structure of KMT5B and of the truncated polypeptides expected from variant 1 and variant 2, respectively. (**b**,**c**) Top view and side view of a model of a nucleosome bound to KMT5B (PDB ID: 8JHG) in ribbon representation (Pymol). In orange: the phosphate backbone. In light blue: the bases of the DNA. In gray: histones. In yellow: truncated protein in the case of the variant Arg187Ter (Patient 2). In pink: lost peptide.

**Table 1 genes-16-01174-t001:** Review of clinical characteristics of patients from the literature and our patients’ clinical findings. ADHD, attention deficit hyperactivity disorder; ASD, autism spectrum disorder; DCD, developmental coordination disorder; ID, intellectual disability; NA, not ascertained.

Patient	Aggregated Data	Percentage	Our Patients
Sex	23 females, 35 males, 6 NA. Total patients 64.	40% female, 58% male	2 females
Age (years) at last evaluation	From 5.5 weeks to 50 years of age	5.5 weeks—50 years (median age around 11 years)	13 years old
Variant type	24 missense variants, 19 frameshift variants, 3 inframe deletions, 2 partial gene deletion, 12 nonsense, 4 splice site	38% missense, 29.5% frameshift, 4.5% inframe deletions, 3% partial gene deletion, 19% nonsense, 6% splice site	Both nonsense variants
Inheritance	45 de novo inherited, 6 inherited variants, 1 plausible germinal mosaicism, 12 NA	86.5% de novo	Both de novo
Macrocephaly	30 macrocephalic patients, 23 normocephalic patients, 11 NA patients	57%	Both macrocephalic
Prominent dysmorphisms and characteristics	45 dysmorphic patients (only either tall or short stature not evaluated), 13 Negative, 6 NA patients	78%	One dysmorphic patient
Global developmental delay	55 GDD, 4 NA/NAP, 5 with speech delay	92%	Both showed GDD
ID/learning difficulties	39 ID/learning difficulties, 3 Negative, 22 NA/NAP	93%	Both shows learning difficulties but no ID (tested)
ADHD	6 ADHD, 49 Negative, 9 NA	11%	Both show ADHD traits
ASD	32 ASD, 17 Negative, 15 NA	65%	Both ASD
Psychosis	2 psychotic patients, 53 Negative, 9 NA/NAP	4%	None
Regression	3 patients showed regression, 54 Negative, 7 NA	5%	None
Behavioral/Psychiatric disorders other than psychosis	12 patients showed behavioral/psychiatric disorders, 42 Negative, 10 NA	22%	No clinically significant disorder
Epilepsy/Seizures	12 Epileptic patients, 40 Negative, 12 NA	23%	None
Sleeping problems	7 patients showed sleeping problems, 50 Negative, 7 NA	12%	Both
Hypotonia	29 hypotonic patients, 21 Negative, 14 NA	58%	None
Movement disorders	6 patients showed a movement disorder (4 motor incoordination/DCD), 59 no, 7 NA	11% (67% motor incoordination/DCD)	Both has DCD
MRI abnormalities	16 patients showed nonspecific findings on MRI, 24 Negative, 24 NA	40% aspecific findings	None
PET/CT brain abnormalities	NA	NA	Both showed abnormalities
Eye abnormalities	13 patients showed eye abnormalities, 38 Negative, 13 NA	22%	None
Congenital anomalies (other than brain anomalies)	13 patients showed congenital anomalies, 48 Negative, 3 NA	21%	None
Others	44 patients showed other clinical problems, 18 Negative, 2 NA	71%	Both showed other clinical signs/symptoms

In Timberlake et al. 2023 [9], the number of patients is not specifiable where clinical signs have been described in one or two patients. In this situation, the highest number of patients showing a particular characteristic is reported, and the rest is considered NA because we cannot know the distribution of signs/symptoms among the three described patients. Furthermore, when a certain diagnosis is age-dependent, as in ID and age of patients is unspecified, patients not showing that phenotype are included among NA.

**Table 2 genes-16-01174-t002:** Functional alterations in two KMT5B case reports. Scores describing an impairment in function are in bold; * scores at the lower normal limits. In the first column, each neuropsychological domain is in gray cells, related sub-domains can be found underneath, in white cells. FSIQ, full-scale intelligence quotient; Zs, Z-score; SS, scaled score; p.le, percentile; Ts, T-score; RS, raw score; WMI, working memory index; PSI, processing speed index; WM, working memory; STM, short-term memory; LTM, long-term memory; TCM, Test delle Campanelle Modificato; ToM, theory of mind. Standardizing scale is dependent on the test used (see methods section for a comprehensive delineation of used tests).

Ascertained Area	Patient 1	Patient 2
Global Cognitive Functioning (WISC IV)	FSIQ 104 (VCI 108, PRI 102, WMI 100, PSI 100)	FSIQ 96 (VCI 100, PRI 119, WMI 76 *, PSI 82 *)
Attention		
Visual Attention (NEPSY-II)	Zs 0.74	Zs 0
Auditory Attention (NEPSY-II)	Zs 0.75	Zs 0.38
TCM	p.le > 10 p.le	**p.le < 10°**
Executive Functions		
Inhibition (Inhibition B NEPSY-II)	SS 11	SS 9
Verbal WM (Forward Verbal Digit Span, Backward Verbal Digit Spa BVN 12–18, Alpha Span MEA)	**Zs -0.29** **Zs 0.43** **p.le < 1°**	**Zs -1.33** **Zs -0.66** **p.le < 1°**
Visuospatial WM (Corsi Block-Tapping Test BVN 12–18)	Zs -0.44	Zs -0.43
Shifting (Inhibition C NEPSY-II)	SS 8	**SS 3**
Planning (TOL)	**Ts 94, Ts > 99, Ts 86** **(total moves, initiation time, and total time in seconds)**	**Ts > 98, Ts 77** **(total moves and rule violations)**
Memory		
Verbal STM (Immediate List Memory NEPSY-II)	SS 12	**SS 5**
Verbal LTM (Delayed List Memory NEPSY-II)	SS 13	SS 6 *
Visuospatial STM (Immediate Memory for Design NEPSY-II)	SS 6 *	**SS 5**
Visuospatial LTM (Delayed Memory for Design NEPSY-II)	SS 12	**SS 5**
Social Perception		
ToM (NEPSY-II)	SS 12	**SS 5**
Emotion Recognition (NEPSY-II)	SS 12	SS 7 *
Sensorimotor Functions		
Imitating Hand Positions (NEPSY-II)	SS 8	**SS 1**
Manual Motor Sequences (NEPSY-II)	SS 8	SS 7 *
VMI	SS 84	SS 84
Language		
Speech Discrimination (BVN 12–18)	Zs 2,16	Zs 2,06
Lexical Denomination (BVN 12–18)	Zs -1.68 *	Zs -0.78
Sentence Generation (BVN 12–18)	**Zs -2.81**	**Zs -3.38**
Emotional and Behavioral Functioning		
Internalizing Problems (CBCL 6–18)	**p.le > 98**	**p.le > 98**
Externalizing Problems (CBCL 6–18)	p.le 84	p.le 89 *
Repetitive Actions or Specific Routines (SCQ-V)	**RS 24 cut-off > 15**	**RS 18 cut-off > 15**
Impaired Communication Skills (SCQ-V)	**RS 24 cut-off > 15**	**RS 18 cut-off > 15**
ADHD Index Conners 3	**Ts 77**	**Ts 70**

## Data Availability

The original contributions presented in this study are included in the article/Appendix A. Further inquiries can be directed to the corresponding author(s).

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
