# Peer review of "Bridging Genotype to Phenotype in KMT5B-Related Syndrome: Evidence from RNA-Seq, 18FDG-PET, Clinical Deep Phenotyping in Two New Cases, and a Literature Review"

_genes, 2025, doi:10.3390/genes16101174_

Round 1

Reviewer 1 Report

Comments and Suggestions for Authors

In the genes-3859744 article, Politano and collaboraters adress the the genotype to phenotype correlation in two female adolescents with  global development delay carrying de novo pathogenic nonsense KMT5B variants. Patients underwent  detailed clinical, neuropsychological, and neuroimaging evaluations, including MRI and 18FDG PET/CT. NGS targeted sequencing and Sanger validation was performed. A targeted gene panel including genes correlated to NDDs was utilized to sequence the DNA of patient 1.  Whole exome sequencing was performed on genomic DNA of patient 2. RNA sequencing was perofrmed to investigate potential differences in gene expression and dysregulated pathways resulting from loss-of-function mutations in KMT5B. Structural effects of both pathogenic variants were analysed. To assess genotype-phenotype correlations, a critical review of the literature was carried out.

Data reported in this manuscript are interesting and well described. Correlation between clinical, deep neuropsychological evaluation and molecular genetic approaches are convencing and well described. The authors' results are validated by the literature data.

Minor comments

  1. KMT5B mutations are subjected to a sex ratio with male predominance. This point would deserve to be mentioned in the manuscript.
  2. Patient 1's deletion is a known variant that has been reported in Gnomad (stop gained, in silico predictor CADD 33). This point should be included in the manuscript.
  3. For the variant c.559C>T; (p.Arg187Ter) identified in patient 2, it is a new variant that has not been reported in the databases. This point deserves to be specified in the manuscript.
  4. Can table 2 be relocated to another part of the manuscript? While the data in the table is relevant to the 2 subjects of the study, he is interrupting the clinical description of patient 1 at the current location. Although it's not of great importance, it would enhance the reading of the manuscript.

Conclusions are in line with the aim objectives of the manuscript

Figures and Table are of good quality

References are complete

Author Response

In the genes-3859744 article, Politano and collaboraters adress the the genotype to phenotype correlation in two female adolescents with  global development delay carrying de novo pathogenic nonsense KMT5B variants. Patients underwent  detailed clinical, neuropsychological, and neuroimaging evaluations, including MRI and 18FDG PET/CT. NGS targeted sequencing and Sanger validation was performed. A targeted gene panel including genes correlated to NDDs was utilized to sequence the DNA of patient 1.  Whole exome sequencing was performed on genomic DNA of patient 2. RNA sequencing was perofrmed to investigate potential differences in gene expression and dysregulated pathways resulting from loss-of-function mutations in KMT5B. Structural effects of both pathogenic variants were analysed. To assess genotype-phenotype correlations, a critical review of the literature was carried out.

Data reported in this manuscript are interesting and well described. Correlation between clinical, deep neuropsychological evaluation and molecular genetic approaches are convencing and well described. The authors' results are validated by the literature data.

Minor comments

  1. KMT5B mutations are subjected to a sex ratio with male predominance. This point would deserve to be mentioned in the manuscript.

We thank the reviewer for this valuable comment and have addressed it accordingly in the Discussion section.

  1. Patient 1's deletion is a known variant that has been reported in Gnomad (stop gained, in silico predictor CADD 33). This point should be included in the manuscript.

We thank the reviewer for the valuable observation; we addressed it accordingly in the Result section.

  1. For the variant c.559C>T; (p.Arg187Ter) identified in patient 2, it is a new variant that has not been reported in the databases. This point deserves to be specified in the manuscript.

We thank the reviewer but the variant has already been described in Patient 9 by Faundes et al. as the reviewer can see in Table 1S.

  1. Can table 2 be relocated to another part of the manuscript? While the data in the table is relevant to the 2 subjects of the study, he is interrupting the clinical description of patient 1 at the current location. Although it's not of great importance, it would enhance the reading of the manuscript.

We thank the reviewer for this observation and agree that the table may not be ideally located. However, according to the Author Guidelines, tables must be placed at the point of their first citation. We are, of course, happy to follow the Reviewer's and Editors’ preference should they wish to relocate it in the manuscript

Conclusions are in line with the aim objectives of the manuscript

Figures and Table are of good quality

References are complete

We thank the Reviewer for this kind and encouraging remarks.

Reviewer 2 Report

Comments and Suggestions for Authors

This paper describes a case report of two individuals diagnosed with KMT5B-Related
Syndrome, thoroughly characterized from the clinical point of view and using a combination of experimental techniques, including imaging and genetic studies. 

The study design is scientifically sound. The methodology and results are adequately described. Conclusions are in line with the results.

Genetic findings are clearly identified and their protein-level implications thoroughly explored.

The authors clearly identify the limitations of the study. 

Author Response

This paper describes a case report of two individuals diagnosed with KMT5B-Related
Syndrome, thoroughly characterized from the clinical point of view and using a combination of experimental techniques, including imaging and genetic studies. 

The study design is scientifically sound. The methodology and results are adequately described. Conclusions are in line with the results.

Genetic findings are clearly identified and their protein-level implications thoroughly explored.

The authors clearly identify the limitations of the study. 

We thank the Reviewer for this kind and encouraging remarks

Reviewer 3 Report

Comments and Suggestions for Authors

The idea of integrating RNA-seq , PET/CT and deep phenotyping is novel and of high interest to the research community. however due to the limited size of samples - the data needs to be cautiously interpreted. 

Minor Comments :

  • Figure 1 : Including arrow etc. to indicate or highlight the features would be helpful.
  • Figure 2: It is hard to interpret this figure.
  • Figure 3 :which regions are mutated ? please highlight 

Major Comments :

  • The sample size of two patients s a limiting factor . Modifying the language to make it more suggestive / indicative  would be important. 
  • While the RNA seq data is great - validating the findings would be necessary . perhaps using cell lines such as sh-sy5y cells may be good models for KD/KO experiments with KMT5B. 
  • The study is missing normal controls. It would help to include them . 
  • Generating EBV immortalized LCLs from the patients in this study would be a good resource  particularly to perform validation experiments  (RNA seq data for example). 

Author Response

The idea of integrating RNA-seq , PET/CT and deep phenotyping is novel and of high interest to the research community. however due to the limited size of samples - the data needs to be cautiously interpreted. 

Minor Comments :

  • Figure 1 : Including arrow etc. to indicate or highlight the features would be helpful.

We thank the reviewer for the valuable comments. We have included the following description in the figure legend: “In the picture the following dysmorphisms can be appreciated: macrocephaly, hypertelorism, downslanting palpebral fissures, horizontal eyebrows, a bulbous nasal tip, and a wide nasal bridge." We generally do not include arrows on clinical photographs, as this may reduce the value of the gestalt appreciation of the facial features.

  • Figure 2: It is hard to interpret this figure.

We thank the reviewer for the valuable comments. We have included the following description in the figure legend: “Images are color-graded from yellow to red, with red indicating stronger differences (higher z-values).” We believe that, by reading the text and examining the figure in light of the caption, the localization of the different areas can now be more accurately appreciated.

  • Figure 3 : which regions are mutated ? please highlight 

We thank the reviewer for the observation. We designed the figure to be as clear as possible: Figure 3a illustrates how the peptide would appear with an early termination caused by p.Arg187Ter and p.Ser832Ter. It then highlights, through the delineation of the p.Arg187Ter-defined boundaries specified in the caption of figure 3b and 3c, the relative localization of the C-terminal and N-terminal regions of the protein with respect to histones and DNA.

Major Comments:

  • The sample size of two patients s a limiting factor . Modifying the language to make it more suggestive / indicative  would be important. 

We thank the reviewer for this specification. We added the sentence “although evidence from more patients is necessary” in the discussion to temper a statement that might have been more assertive than warranted. Additionally, we modified the conclusions to read: “Future studies should aim to recruit a larger cohort of patients to strengthen our PET and RNA-seq findings and their correlation with neuropsychiatric symptoms and signs, not only in KMT5B-related syndrome but also in other genetically confirmed complex neurodevelopmental disorders. The main limitations of our study include the small sample size, due to the rarity of the disease, and the genetic heterogeneity between the two patients, despite both mutations likely resulting in a loss-of-function (LoF) effect.”

  • While the RNA seq data is great - validating the findings would be necessary . perhaps using cell lines such as sh-sy5y cells may be good models for KD/KO experiments with KMT5B. 

We thank the reviewer for the valuable observation. We agree that validation in model systems would be valuable; however, we believe this represents a separate set of experiments that could be addressed in future studies. Importantly, the observation that both patients, carrying distinct KMT5B variants, exhibit highly similar RNA-seq profiles provides a form of preliminary internal validation and supports the robustness of our findings. We believe this consistency already offers meaningful insight while remaining aligned with the scope of the current article.

  • The study is missing normal controls. It would help to include them . 

In the Methods section, we specified that age- and sex-matched normal controls were used for both the PET study and the RNA-seq analysis. All statistical analyses were performed by comparison with these normal controls, and therefore we believe that this point has been thoroughly addressed in the Methods section.

  • Generating EBV immortalized LCLs from the patients in this study would be a good resource  particularly to perform validation experiments  (RNA seq data for example). 

We thank the reviewer for the observation. We agree that further functional validation would be valuable to better understand KMT5B-related disease. However, such experiments would require substantial additional work and fall beyond the primary focus of the present study. Importantly, the observation that both patients, carrying distinct KMT5B variants, exhibit highly similar RNA-seq profiles provides a form of preliminary internal validation and supports the robustness of our findings. We believe this consistency already offers meaningful insight while remaining aligned with the scope of the current article.
